# PDE-Diffusion: Physics guided diffusion model for solving partial differential equations

## Abstract

Solving partial differential equations (PDEs) is crucial in various disciplines, and their resolution often necessitates the use of computationally intensive numerical methods as well as specialized domain expertise. While data-driven approaches have emerged as promising alternatives, they encounter limitations in terms of generalizability, interpretability, and long-horizon predictive performance, as well as issues related to temporal incoherence. To address these challenges, we introduce the PDE-Diffusion, a two-stage model with three distinctive features: (i) the incorporation of physics-based priors to enhance model interpretability and generalization, (ii) a two-stage diffusion model that efficiently handles physical field forecasting without requiring multi-frame inputs, and (iii) the assimilation of PDE-informed constraints to ensure temporal coherence while producing high-quality predictive results. We conduct extensive experiments to evaluate PDE-Diffusion's capabilities using the PDEBench dataset and two of our newly proposed datasets. The results indicate that PDE-Diffusion delivers state-of-the-art performance in all cases.

## 1 Introduction

Solving partial differential equations (PDEs) is a critical task across various domains in physical science, engineering, and biology (Gao et al., 2022). In the field of physics, the dynamics of physical fields are often governed by PDEs, such as the Navier-Stokes equations and Maxwell equations. Traditional numerical methods like the Finite Element Method (FEM) (Guptaapos; & Meek', 1996) address the problem by discretizing the physical field onto a computational grid. This is typically followed by months-long simulations utilizing iterative numerical solvers. Despite their effectiveness, these conventional techniques require significant computational resources. Moreover, they necessitate recalculations when confronted with new initial conditions, boundary conditions, or different PDE systems (Li et al., 2020a).

As deep learning has been a promising tool in various fields, including computer vision and natural language processing, it also provides a new way to solve PDEs rapidly by learning from a large amount of observed data (Raissi et al., 2019). Such techniques have demonstrated success in several applications, including physical field forecasting (Raissi et al., 2017a) and data-driven discovery of a PDE system's intrinsic physical laws (Raissi & Karniadakis, 2018). To mitigate the limitations of traditional numerical methods, (Raissi et al., 2017b) introduced Physics-Informed Neural Networks (PINNs), a deep learning (DL)-based alternative for solving PDEs. This approach offers a means to integrate prior physical knowledge into the model through regularization techniques. However, despite the advantages of computational efficiency and the innovative framework for embedding prior physical knowledge, current deep-learning-based PDE solvers often suffer from limitations in generalization across different initial and boundary conditions (Kovachki et al., 2021).

Subsequent advancements in the field of neural operators (Li et al., 2020b;a; Lu et al., 2021; Bhattacharya et al., 2021; Patel et al., 2021) have signaled a paradigm shift toward employing mesh-free, infinite-dimensional operators through neural networks. This approach offers distinct advantages for simulating the complex dynamics of nonlinear systems under varying boundary and initial conditions, all without necessitating retraining of the neural network (Higgins, 2021). Despite their

capabilities, these operators are primarily limited to making single-step field predictions and are not optimally configured for multi-step forecasting tasks, namely, one-to-many prediction. This limitation deviates from the conventional problem formulation in PDE solving. Moreover, existing transformer-based and Recurrent Neural Network (RNN)-based models are equally inappropriate for these specific tasks. The former encounter difficulties in handling multi-step forecasting, whereas the latter are frequently compromised by the accumulation of errors. As a result, there exists an imperative to devise a novel architectural approach that can adeptly tackle physical field multi-step forecasting.

In comparison, Generative diffusion models offer substantial potential for solving PDEs, exhibiting capabilities similar to those required in video generation based on initial frames and auxiliary conditioning (Rombach et al., 2021). Specifically, diffusion models excel at generating high-resolution outputs, a feature highly beneficial for PDE simulations. Originally developed for image generation in computer vision (Rombach et al., 2022; Dhariwal & Nichol, 2021b), these models have also been extended to address challenges in natural language processing (Austin et al., 2023; He et al., 2022) and have also demonstrated efficacy in multi-modal tasks (Kim et al., 2022). By constructing generative distributions that closely approximate the underlying data distributions, these models offer robust generalization capabilities, a feature highly desirable for modeling intricate PDE systems. However, despite their apparent suitability for PDE system modeling, generative models have seen limited adoption in this domain. This is mainly due to a lack of scientific priors guidance during the reverse generative process. Thus, the design of physics-embedded generative models for highly accurate predictions in PDE systems remains a largely unexplored avenue in the field of AI for science.

To address the challenges outlined above, we introduce the **PDE-Diffusion**, a specialized generative diffusion model designed for physical field forecasting. This model integrates our newly-proposed, interpretable, physics-embedded framework to effectively utilize predefined physical properties of PDEs, initial conditions (ICs), boundary conditions (BCs), and specific PDE types. Our framework comprises three key components: (1) BCs Embedding, which employs a masking strategy to the physical field and inject it along with ICs embedding into the diffusion model during the reverse process, thereby endowing the network with the ability to generalize and adaptively forecast new physical fields with distinct ICs and BCs; (2) PDE Type Embedding, which serves as an additional physical constraint to guide the model's understanding of general PDE dynamics; and (3) Latent Residual Prediction Strategy, which leverages PDE-informed constraints to address the issue of temporal incoherence in physical fields. We evaluate the effectiveness of PDE-Diffusion using the established PDEBench dataset (Takamoto et al., 2023b), a leading benchmark for PDE system simulation tasks, as well as two additional datasets that we have created. Our results establish PDE-Diffusion as a state-of-the-art model for solving PDEs, thereby setting a new benchmark in the field. The main contributions of this paper are summarized as follows:

- We introduce PDE-Diffusion, a generative model designed for one-to-many physical field predictions in PDE solving, offering greater interpretability and generalizability.

- We have developed a physics-embedded framework for the diffusion process by incorporating predefined physical constraints of PDEs for controlled generation. This confirms the architecture's ability to effectively capture the essential physics governed by PDEs. It makes the model more interpretable and allows the pretrained model to generalize to physical fields with new ICs, BCs, or even PDE types. A notable feature is the inclusion of a latent residual to mitigate the problem of temporal incoherence in physical field predictions.

- We introduce two new simulated datasets designed to emulate real-world challenges. This enhances the available resources for deep learning in the PDE-solving domain. These datasets specifically tackle the shortcomings associated with the limited availability of 2D PDE datasets.

## 2 RELATED WORKS

**Denoising Diffusion Models.** Diffusion models, as a specialized class of generative probabilistic models, were first explored in a seminal work (Sohl-Dickstein et al., 2015). Subsequently, (Ho et al., 2020b) introduced a framework employing both forward noisy processes and backward denoising processes for sample generation based on empirical data distributions. (Song et al., 2020a) further

proposed DDIM, which accelerates sampling by using subsequence sampling and achieves faster sampling with fewer steps. In a subsequent development, (Dhariwal & Nichol, 2021c) introduced a classifier that guides the sampling process everaging gradients of the target class. Additionally, improvements were made to the model's backbone by incorporating self-attention structures, resulting in higher quality and diversity compared to images generated by GANs, as well as more stable training. Most recently, (Liu et al., 2022) improved the guiding methods by introducing text-guided, image-guided, and mixed-guided approaches.

**Diffusion Models for Video Modeling.** Given the impressive performance of diffusion models in image processing, diffusion models have now been extended to the realm of video modeling. (Yang et al., 2023) introduced video diffusion models as a natural progression from standard image architectures. These models have been further refined for various types of video prediction tasks. For instance, (Höppe et al., 2022) presented Random-Mask Video Diffusion (RaMViD), which employs 3D convolutions to extend image diffusion models to video prediction and infilling, incorporating a novel conditioning technique during the training phase. However, such models often falter in capturing global temporal coherence, relying merely on conditions from a few previous frames, which can result in inconsistent or even erroneous outcomes in long-term video predictions. To address this, (Yang et al., 2023) proposed the Local-Global Context-guided Video Diffusion model (LGC-VD), which captures multi-perception conditions to produce high-quality videos in both conditional and unconditional settings. Building upon these advancements, (Ni et al., 2023) introduced Latent Flow Diffusion Models (LFDM), by employing a Conditional Image-to-Video (cI2V) framework, LFDM excels in synthesizing complex spatial details and temporally coherent motion patterns by fully leveraging the spatial content inherent in the input images. However, despite significant advancements in video modeling with diffusion models, the literature still reveals a gap, particularly in the area of integrating physics principles from PDEs for the purpose of physical field forecasting.

**Deep Learning Architectures for Solving PDEs.** Physics-Informed Neural Networks (PINNs) introduced by (Raissi et al., 2019), serve as the first physical knowledge guided deep learning (DL)-based alternative for solving Partial Differential Equations (PDEs). Nonetheless, they are recognized as a specialized form of neural Finite Element Methods (FEMs)(Li et al., 2020a). For a more comprehensive utilization of the physical information inherent in PDEs, (Guen & Thome, 2020) developed PhyDNet that employs a dual-branch architecture inspired by data assimilation techniques. While these networks offer mesh-independent and precise solutions, they necessitate retraining for new instances of different PDE parameters, such as variations in initial and boundary conditions (ICs, BCs). To address the issues of retraining requirements, DeepOnet was introduced by (Lu et al., 2021). Unlike traditional neural FEM that learn mappings between finite-dimensional Euclidean spaces, it directly learns the functional relationship between parametric dependencies and their corresponding solutions. Then (Li et al., 2022) proposed Fourier Neural Operators (FNO), which parameterize the integral kernel directly in Fourier space, thereby enabling both expressive and efficient architectures. Nevertheless, FNO falls short in the active integration of physical principles from PDEs. Additionally, the constraints imposed through PINNs tend to be overly stringent, thereby limiting the model's adaptability to new initial and boundary conditions (ICs, BCs). To address these limitations, we present PDE-Diffusion, which employs a novel physics-embedding framework to guide the diffusion process, This approach takes into account well-defined ICs, BCs, and specific types of PDEs, thereby capturing the intrinsic dynamics of the PDE system effectively.

## 3 PRELIMINARIES

### 3.1 NOTATIONS AND PROBLEM DEFINITION

**Notations.** Given a vector field $U(x, t) \in \mathbb{R}^C$, where $x, t$ are the spatial and temporal coordinates, respectively, defined on a discrete grid: $x \in \Omega = \{x^1, x^2, \ldots, x^M\}, t \in I = \{t^1, t^2, \ldots, t^T\}$. Here, $U$ represents a vector field comprising $C$ physical variables (velocity, pressure, temperature , etc). The governing partial differential equations (PDE) for this system can be represented as:

$$\frac{\partial U}{\partial t} = \mathcal{S}(U, x, t), \tag{1}$$

where $\mathcal{S}$ stands for the temporal differentiation of the field.

**Problem Definition.** The *PDE solving problem* can be formulated as to find the exact solution throught the grid $I \times \Omega$, given the initial condition $u^0 = U(x, t^0), \forall x \in \Omega$ and boundary condition $u^B = U(x, t), \forall (x, t) \in \partial \Omega \times I$.

Therefore within the simplest first-order time discretization scheme, we have:

$$u^{i+1} \approx (t^{i+1} - t^i)\mathcal{S}(u^i, x, t) + u^i,\qquad(2)$$

For simplicity, we denote $u^i = U(x, t_i)$ the vector representation of the field at time step $i$.

## 4 METHOD

In contrast to conventional Computational Fluid Dynamics (CFD) algorithms, which require a carefully designed and computationally expensive scheme to determine $\mathcal{S}(u^i, x, t)$, this paper introduces PDE-Diffusion. We cast the problem of physical field forecasting as a spatiotemporal sequence prediction task. PDE-Diffusion operates as a two-stage model, harnessing the physics-based information of PDEs for physical field forecasting. The first stage comprises an autoencoder for feature extraction, while the second stage deploys a diffusion model for predictive tasks. The two training stages of PDE-Diffusion are illustrated in Fig 1. A detailed exposition of our two-stage framework is provided in the subsequent sections.

### 4.1 LATENT REPRESENTATION OF VECTOR FIELD

In the first stage, we train an autoencoder to compress the input field $u^i$ into its latent representation $z^i$, as illustrated in Fig. 1. Specifically, during each iteration, we randomly select a minibatch of physical fields $u^i$, each with dimensions $[H, W, C]$ at time-step $i$, where $C$ represents the number of physical variables comprised in the physical field. The encoder $\mathcal{E}$ then maps each input field $u^i$ to its corresponding latent representation $z^i$, which has dimensions $[H_z, W_z, C_z]$. Subsequently, the decoder $\mathcal{D}$ transforms the latent vector $z^i$ back into the reconstructed field $\hat{u}^i = \mathcal{D}(\mathcal{E}(u^i))$. The training algorithm for this process is presented in Algorithm 1. Optimization of the autoencoder is conducted in an unsupervised manner.

$$\min_{\theta_{\mathcal{E}}, \theta_{\mathcal{D}}} \sum_{i \in [1, N]} \| u^i - \hat{u}^i \|,\qquad(3)$$

where $N$ is the number of physical fields in one minibatch.

### 4.2 PHYSICS-EMBEDDED DIFFUSION MODEL

Then in the second stage, the diffusion model with physical information embedded is trained to generate the latent residual vector $d^i$ at each time step, so that the decoder can reconstruct the entire physical field for all subsequent time steps based solely on the initial conditions $u_0$ and the boundary condition $u_B$. The predicted physical field can be simply obtained by : $\hat{u}^i = \mathbf{D}(z^0 + d^i)$. The PDE-Diffusion can be formulated as $N$ independent Markov chains : $d^i = (d_1^i, \ldots, d_n^i, \ldots d_N^i), i \in [1, T]$. The subscript $n$ represents the sampling step of DDPM, while the superscript $i$ represents the time index. During training, we first sample a full clip of physical field $u_0 = (u_0^1, \ldots, u_0^i, \ldots u_0^T)$ from the data distribution, and then use the encoder $\mathcal{E}$ to get $d_0 = \mathcal{E}(u_0)$ . Next, we perform the forward process of DDPM by continuously adding Gaussian noise according to the variance schedule $\{\beta_n\}, n \in [1, N]$ :

$$q(d_{n+1}^i | d_n^i) = \mathcal{N}(d_{n+1}^i; \sqrt{1 - \beta_n} d_n^i, \beta_{n+1}\mathbf{I})\qquad(4)$$

The reverse process $q(d_n^i | d_{n+1}^i)$ can be estimated by:

$$p_{\theta_{DM}, \phi}(d_n^i | d_{n+1}^i, e) = Z \cdot p_{\theta_{DM}}(d_n^i | d_{n+1}^i) p_\phi(e | d_n^i).\qquad(5)$$

Here $p_\phi$ measure the probability that $d_n^i$ conforms to the governing equations $e$. $Z$ is a normalizing constant(Dhariwal & Nichol, 2021a). Recall that in equation1, the increment of the physical field can be calculated by $\mathcal{S}(U, x, t)$. Therefore after sampling $d_n$ , the decoder $\mathcal{D}$ pretrained in the first

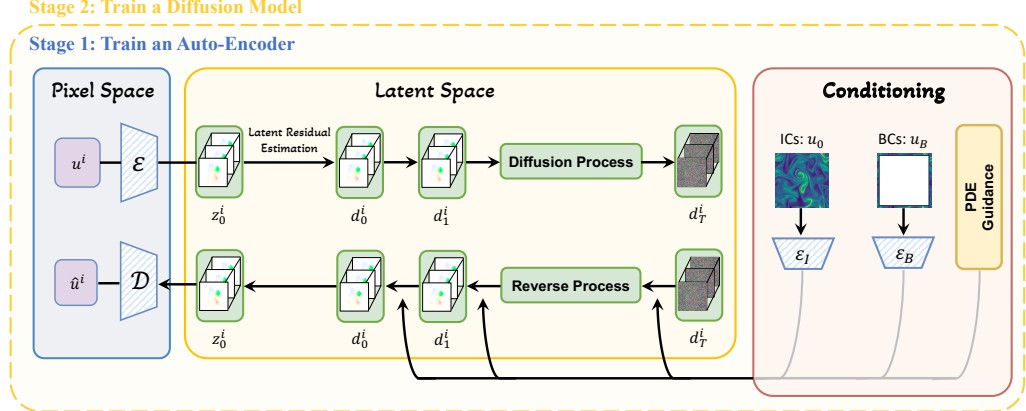

Figure 1: Training stages of PDE-Diffusion. In the stage 1, we employ an Auto-Encoder, which comprises two trainable modules: an encoder ($\mathcal{E}$) to map the input field $u^i$ into its latent representation, and a decoder ($\mathcal{D}$) to reconstruct the physical field $\hat{u}^i$ from the latent vectors. During the stage 2, we leverage the pre-trained Auto-Encoder from stage one. A collection of input physical fields, each denoted as $u^i$, is fed into $\mathcal{E}$. Subsequently, we estimate the latent residuals between each pair of $u^i$ fields and dispatch these residuals for diffusion. Throughout the reverse process, at each time step, a conditioning module incorporates the physical constraints associated with PDEs, ICs, and BCs. Additionally, a PDE guidance is employed to leverage the specific type of PDE, thereby guiding the model in the prediction of physical fields.

stage is used to reconstruct the noisy flow field $\hat{u}_n = (\hat{u}_n^1, \hat{u}_n^2, \ldots, \hat{u}_n^T)$ at sampling step $n$. Hence we can define $p_\phi$ as:

$$p_\phi(e|d_n^i) = \exp\left(- \parallel \mathcal{E}(\hat{u}_n^i) - \mathcal{E}(\hat{u}_n^{i-1} + (t^i - t^{i-1})\mathcal{S}(\hat{u}_n^{i-1}, x, t)) \parallel\right) \tag{6}$$

$$= \exp\left(- \parallel \mathcal{E}(\hat{u}_n^i) - C \parallel\right) \tag{7}$$

$$= \exp\left(- \parallel \epsilon_\phi(d_n^i) \parallel\right), \tag{8}$$

where $\mathcal{E}(\hat{u}_n^{i-1} + (t^i - t^{i-1})\mathcal{S}(\hat{u}_n^{i-1}, x, t))$ can be simplified as a constant $C$ cause $\hat{u}_i^n$ and $\hat{u}_{i-1}^n$ are independent. We use a spatial derivative kernel to approximate $\mathcal{S}(\hat{u}_n^{i-1}, x, t)$ to accelerate training. Discretization scheme with higher order can also be employed to trade off speed with accuracy. Compared to DM for classic CV tasks, PDE-Diffusion requires more determinacy. To this end, we use the score-based conditioning trick proposed in (Song et al., 2020b), so that the reverse process can be processed in a deterministic DDIM manner. In particular, PDE-Diffusion predicts the error term based on the score function:

$$\nabla_{d_n^i} \log\left(p_{\theta_{DM}}(d_n^i|d_{n+1}^i)p_\phi(e|d_n^i)\right) = \nabla_{d_n^i} \log p_{\theta_{DM}}(d_n^i|d_{n+1}^i) + \nabla_{d_n^i} \log p_\phi(e|d_n^i) \tag{9}$$

$$= -\frac{1}{\sqrt{1 - \bar{\alpha}_n}}\epsilon_{\theta_{DM}}(d_n^i) - 2\epsilon_\phi(d_n^i), \tag{10}$$

where $\bar{\alpha}_n = \prod_{s=0}^n (1 - \beta_s)$ (Ho et al., 2020a). Now we can define the final expression of error term:

$$\hat{\epsilon}_{\theta_{DM}, \phi}(d_n^i) := \epsilon_{\theta_{DM}}(d_n^i) + s_n \epsilon_\phi(d_n^i), \tag{11}$$

where $s_n = 2\sqrt{1 - \bar{\alpha}_n}$ is a set of constant scale that controls the degree of physics embedding $\epsilon_\phi(d_n^i)$. A detailed prof of 9 is provided in appendix. To denoise $d_n^i$, a 3D UNet is trained to estimate the noise term $\epsilon_{\theta_{DM}}$ at step $n$. The parameters $\phi$ including autoencoder and derivative kernel are fixed in this stage. The training process is to minimize:

$$\min_{\theta_{DM}} \mathbb{E}_{n \sim \mathcal{U}(1,N), d_0^i \sim q(d_0^i), \epsilon \sim \mathcal{N}(\mathbf{0}, \mathbf{I})}[\parallel \hat{\epsilon}_{\theta_{DM}, \phi}(d_n^i) - \epsilon \parallel]. \tag{12}$$

The complete algorithm, including the two training stages, is shown in Algorithm 1 and Algorithm 2. The reverse sampling algorithm is presented in the Appendix as Algorithm 3.

---

**Algorithm 1** Training Stage 1: Train the Auto-Encoder (AE)

---

1: **Initialize:** Encoder $\mathcal{E}$ and decoder $\mathcal{D}$ with parameters $\theta_{\mathcal{E}}$ and $\theta_{\mathcal{D}}$
2: **for** each training iteration **do**                                          ▷ Training Loop
3:      Randomly select a minibatch of physical fields $u^i$, each with dimensions $[H, W, C]$
4:      $z^i = \mathcal{E}(u^i)$ with dimensions $[H_z, W_z, C_z]$       ▷ Encode $u^i$ into latent representation $z^i$
5:      $\hat{u}^i = \mathcal{D}(z^i)$ with dimensions $[H, W, C]$       ▷ Reconstruct $\hat{u}^i$ from the latent vector
6:      **Objective Function:**                                          ▷ Optimization Step

$$\min_{\theta_{\mathcal{E}}, \theta_{\mathcal{D}}} \sum_{i \in [1,N]} \| u^i - \hat{u}^i \|$$

7: **end for**

---

---

**Algorithm 2** Training Stage 2: Train the Diffusion Model

---

1: **Initialize:** Diffusion model parameters $\theta_{DM}, \phi$
2: **for** each training iteration **do**                                          ▷ Training loop
3:      $d_0^i \sim q\left(d_0^i\right)$                                          ▷ Sample initial latent residual vector $d_0^i$
4:      $t \sim \text{Uniform}\{1, \ldots, T\}$                                          ▷ Sample $t$
5:      $\epsilon \sim \mathcal{N}(\mathbf{0}, \mathbf{I})$                                          ▷ Sample noise $\epsilon$
6:      Obtain a $d_n^i$ by adding noise according to the variance schedule $\{\beta_n\}$
7:      Update $\theta_{DM}$ to minimize the objective function:

$$\min_{\theta_{DM}} \mathbb{E}_{n \sim \mathcal{U}(1,N), d_0^i \sim q(d_0^i), \epsilon \sim \mathcal{N}(\mathbf{0}, \mathbf{I})} [\| \hat{\epsilon}_{\theta_{DM}, \phi}(d_n^i) - \epsilon \|]$$

8: **end for**

---

## 5 EXPERIMENTS

### 5.1 DATASETS

We adopt 4 datasets including CFD2D, DR2D, Double Mach and RMI, which provide data of all the physical fields appeared in their partial differential equation. We provide detailed descriptions of the four datasets in appendix B.

**CFD2D** is a simulation dataset of time-dependent compressible flow in 2D space. It contains eight different sub-cases based on the initial condition setting and PDE coefficients. CFD2D is generated using periodic boundary condition on a $512 \times 512$ grid.
**DR2D** (diffusion-reaction 2D) provides 2D simulation data of the diffusion-reaction system. It has only two non-linearly coupled variables, namely the activator and the inhibitor, which is commonly used in the field of chemistry.

We also provides two new datasets to enrich the PDE solving benchmarking:

**Double Mach** provides a standard test case for Euler solvers which is widely used in testing CFD algorithms. We use WENO5 scheme to generate this dataset to simulate the evolution of a tilted shock wave on the left sige of the flow field with a Mach number of 10.
**RMI** (Richtmeyr-Meshkov Instability) dataset describes a flow system in which the low-density fluids accelerate high-density fluids in the gravity field. We generate 10 cases with different gravitational acceleration to simulate the Rayleigh Taylor instability phenomenon.

### 5.2 EXPERIMENTAL DETAILS

#### 5.2.1 BASELINES

We compare PDE-Diffusion with three baseline models, including: FNO (Li et al., 2020a), UNet (Ronneberger et al., 2015), and Video Diffusion Model (VDM) (Ho et al., 2022). The VDM is adapted to the PDE solving task by simply replace the text conditioning with encoded initial condition and boundary condition. The training process of VDM remains the same.

Table 1: Partial derivative equations solving results on 4 datasets (5 cases) with PDE-Diffusion, FNO, UNet and VDM.

| Methods | | PDE-Diffusion | | FNO | | Unet | | VDM | |
|---|---|---|---|---|---|---|---|---|---|
| Metric | | MSE | MAE | MSE | MAE | MSE | MAE | MSE | MAE |
| Double Mach | 2 | **0.113** | **0.135** | 0.146 | 0.158 | 0.303 | 0.209 | 0.245 | 0.217 |
| | 8 | **0.336** | **0.367** | 0.928 | 0.498 | 0.496 | 0.327 | 0.456 | 0.458 |
| | 16 | **0.409** | **0.288** | 3.142 | 1.202 | 0.994 | 0.524 | 0.887 | 0.654 |
| RMI | 2 | 0.335 | 0.359 | 0.168 | 0.173 | **0.067** | **0.100** | 0.439 | 0.423 |
| | 8 | **0.308** | **0.324** | 0.534 | 0.718 | 0.958 | 0.554 | 0.673 | 0.701 |
| | 16 | **0.781** | **0.562** | 2.764 | 1.047 | 1.583 | 0.588 | 1.766 | 0.798 |
| DR2D | 2 | 0.024 | 0.057 | 0.004 | 0.025 | 0.009 | 0.072 | 0.146 | 0.235 |
| | 8 | **0.038** | **0.112** | 0.056 | 0.190 | 0.151 | 0.289 | 0.456 | 0.458 |
| | 16 | **0.087** | **0.275** | 0.194 | 0.345 | 0.318 | 0.416 | 0.581 | 0.634 |
| CFD2D turb1 | 2 | 0.003 | 0.210 | **3e-4** | **0.163** | 0.076 | 0.185 | 0.039 | 0.223 |
| | 8 | **0.015** | **0.113** | 0.029 | 0.165 | 0.465 | 0.317 | 0.188 | 0.258 |
| | 16 | **0.123** | **0.229** | 0.229 | 0.364 | 1.786 | 1.261 | 0.675 | 0.554 |
| CFD2D turb2 | 2 | **0.012** | **0.113** | 0.022 | 0.168 | 0.176 | 0.251 | 0.539 | 0.523 |
| | 8 | **0.278** | **0.248** | 0.395 | 0.271 | 0.419 | 0.451 | 0.756 | 0.658 |
| | 16 | **0.768** | **0.681** | 0.656 | 0.537 | 1.471 | 0.935 | 1.213 | 1.084 |

### 5.2.2 HYPERPARAMTERS

The autoencoder in PDE-Diffusion follows the same architecture in (Johnson et al., 2016), with 2 down-sampling blocks, a bottleneck with 6 residual blocks, and 2 upsampling blocks. We train the autoencoder for 300 epoch with a learning rate $4 \times 10^{-4}$ that decay 2 time smaller every 80 epochs. In the second stage, the diffusion model uses 3D-UNet (Ho et al., 2022) as denoising function, which includes 4 down-sampling and 4 up-sampling 3D convolutional blocks. Both of the model is trained using Adam optimizer (Kingma & Ba, 2014).
**Setup:** The input of each dataset is zero-mean normalized. To better explore the PDE guidance's effectiveness in our proposed PDE-Diffusion, we evaluate PDE-Diffusion on two different prediction window size: 8 and 16. FNO and UNet are firstly trained to predict one step ahead and then predict the subsequent physical field in a autoregressive manner.
**Platform**: the PDE-Diffusion is trained and tested on two Nvidia RTX 4090 24GB GPUs with a batch size of 8.

### 5.3 RESULT ANALYSIS

Table 2 and table 3 summarize the experimental results on 4 datasets (11 cases). We can observe that: (1) The proposed PDE-Diffusion significantly improve the performance of solving PDEs across almost every cases. Specifically, PDE-Diffusion has a MSE decrease of 21% compared to FNO. (2) By introducing the PDE guidance, PDE-Diffusion has a decrease of x% in average in the PRE metric compared to other baseline models. Considered that PDE-Diffusion share the same 3D-UNet denoising model architecture with VDM, the notable improvement provides strong evidence for the effectiveness of physical embedding in PDE-Diffusion. (3) Our method shows significant better results in the case of longer prediction window size. The MSE decrease is x% (at 2), x% (at 4), x% (at 8), x% (at 16), respectively, which proves that the PDE guidance and residual forecasting can effectively increase the stability in long sequence prediction.

### 5.3.1 CASE STUDY

As shown in figure 2, FNO exhibits relatively accurate predictions in the early steps, but shows a sharply deteriorates of performance in long-term forecasting. Compared to FNO, PDE-Diffusion can better capture the transition patterns of the flow field and provides clearer results at fluid interfaces with fewer distortions. We also observed that PDE-Diffusion's results show a significantly better distinction between various regions in the Double Mach test case compared to other baselines.

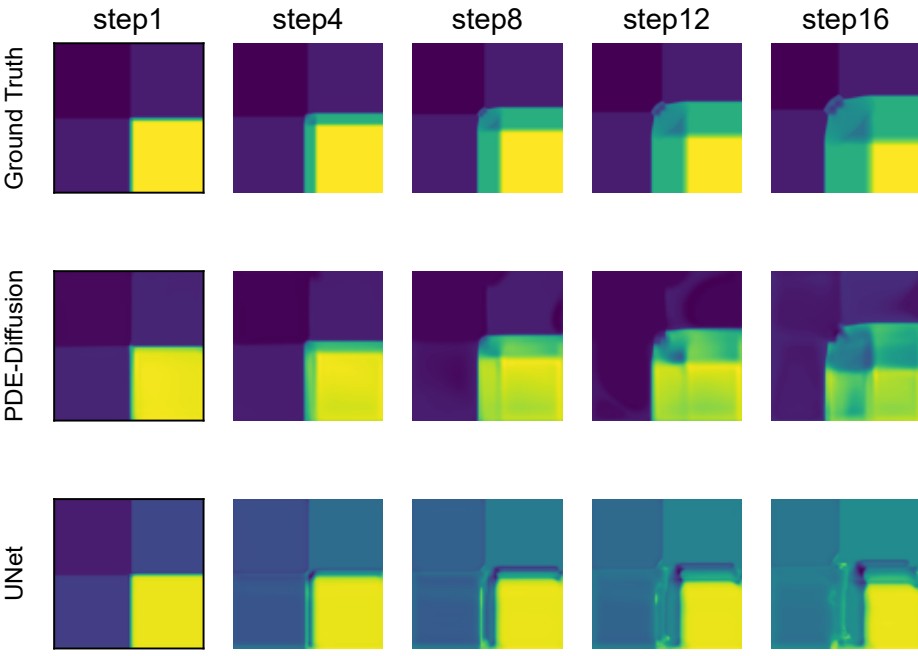

Figure 2: PDE-Diffusion and Unet's density field prediction results on the Double Shock dataset over 1, 2, 4, 8, and 16 autoregressive steps, with the first row being ground truth, the second row PDE-Diffusion, and the third row Unet.

Table 2: Partial derivative equations solving results on 4 datasets (5 cases) with PDE-Diffusion, FNO, UNet and VDM.

| Methods | | PDE-Diffusion(ours) | | FNO | | Unet | | VDM | |
|---|---|---|---|---|---|---|---|---|---|
| Metric | | MSE | MAE | MSE | MAE | MSE | MAE | MSE | MAE |
| Double Mach | 2 | **0.402** | **0.448** | 0.146 | 0.158 | 0.303 | 0.209 | 0.439 | 0.423 |
| | 8 | **0.478** | **0.498** | 0.928 | 0.498 | 0.496 | 0.327 | **0.456** | 0.458 |
| | 16 | **0.768** | **0.681** | 3.142 | 1.202 | 0.994 | 0.524 | 0.591 | 0.684 |
| RMI | 2 | **0.402** | **0.448** | 0.168 | 0.173 | 0.027 | 0.100 | 0.439 | 0.423 |
| | 8 | **0.478** | **0.498** | 0.534 | 0.718 | 1.66 | 0.554 | **0.456** | 0.458 |
| | 16 | **0.768** | **0.681** | 2.764 | 1.047 | 1.583 | 0.588 | 0.591 | 0.684 |
| DR2D | 2 | **0.402** | **0.448** | 0.004 | 0.025 | 0.009 | 0.072 | 0.439 | 0.423 |
| | 8 | **0.478** | **0.498** | 0.056 | 0.190 | 0.151 | 0.289 | **0.456** | 0.458 |
| | 16 | **0.768** | **0.681** | 0.194 | 0.345 | 0.318 | 0.416 | 0.591 | 0.684 |
| CFD2D turb1 | 2 | **0.402** | **0.448** | 3e-4 | 0.163 | 32.76 | 6.155 | 0.439 | 0.423 |
| | 8 | **0.478** | **0.498** | 0.029 | 0.165 | 60.465 | 6.117 | **0.456** | 0.458 |
| | 16 | **0.768** | **0.681** | 0.029 | 0.164 | 63.786 | 6.261 | 0.591 | 0.684 |
| CFD2D turb2 | 2 | **0.402** | **0.448** | 0.022 | | 0.576 | 0.951 | 0.439 | 0.423 |
| | 8 | **0.478** | **0.498** | 5.395 | 1.671 | 1.519 | 0.951 | **0.456** | 0.458 |
| | 16 | **0.768** | **0.681** | 23.156 | 2.937 | 1.471 | 0.935 | 0.591 | 0.684 |

## 5.4 ABLATION STUDY

We also design additional experiments on Double Mach with ablation consideration. We remove respectively BC embedding, PDE guidance and latent residual prediction strategy to assess their impact on the PDE-Diffusion. The experimental results are shown in table 4. We observe a 37% of MSE decrease without PDE guidance. The results demonstrate that PDE guidance contributes the most to PDE-Diffusion, while the A strategy significantly reduces the affine problem in the physical field. The introduction of BC embedding effectively enhances the numerical stability of the physical field at the boundaries.

Table 3: Partial derivative equations solving results on CFD2D dataset (6 random initialized cases) with PEDM, FNO, UNet and VDM

| Methods | | PEDM(ours) | | FNO | | Unet | | VDM | |
|---|---|---|---|---|---|---|---|---|---|
| Metric | | MSE | MAE | MSE | MAE | MSE | MAE | MSE | MAE |
| CFD2D rand1 | 2 | **0.402** | **0.448** | 5e-4 | 0.028 | 0.006 | 0.073 | 0.439 | 0.423 |
| | 8 | **0.478** | **0.498** | 1.268 | 0.501 | 0.721 | 0.415 | **0.456** | 0.458 |
| | 16 | **0.768** | **0.681** | 1.268 | 0.501 | 0.815 | 0.440 | 0.591 | 0.684 |
| CFD2D rand2 | 2 | **0.402** | **0.448** | 0.016 | 0.032 | 2.021 | 1.284 | 0.439 | 0.423 |
| | 8 | **0.478** | **0.498** | 1.170 | 0.528 | 6.267 | 2.406 | **0.456** | 0.458 |
| | 16 | **0.768** | **0.681** | 1.168 | 0.529 | 9.643 | 2.504 | 0.591 | 0.684 |
| CFD2D rand3 | 2 | **0.402** | **0.448** | 0.012 | 0.018 | 0.827 | 0.452 | 0.439 | 0.423 |
| | 8 | **0.478** | **0.498** | 1.232 | 0.454 | 1.014 | 0.486 | **0.456** | 0.458 |
| | 16 | **0.768** | **0.681** | 1.172 | 0.453 | 1.025 | 0.490 | 0.591 | 0.684 |
| CFD2D rand4 | 2 | **0.402** | **0.448** | 0.012 | 0.021 | 0.009 | 0.072 | 0.439 | 0.423 |
| | 8 | **0.478** | **0.498** | 1.172 | 0.453 | 0.151 | 0.289 | **0.456** | 0.458 |
| | 16 | **0.768** | **0.681** | 1.172 | 0.453 | 0.318 | 0.416 | 0.591 | 0.684 |
| CFD2D rand5 | 2 | **0.402** | **0.448** | 0.248 | 0.121 | 0.539 | 0.304 | 0.439 | 0.423 |
| | 8 | **0.478** | **0.498** | 1.157 | 0.489 | 0.991 | 0.628 | **0.456** | 0.458 |
| | 16 | **0.768** | **0.681** | 1.153 | 0.461 | 0.997 | 0.631 | 0.591 | 0.684 |
| CFD2D rand6 | 2 | **0.402** | **0.448** | 0.002 | 0.249 | 0.894 | 0.517 | 0.439 | 0.423 |
| | 8 | **0.478** | **0.498** | 1.211 | 0.485 | 1.354 | 0.778 | **0.456** | 0.458 |
| | 16 | **0.768** | **0.681** | 1.211 | 0.485 | 1.392 | 0.792 | 0.591 | 0.684 |

Table 4: Ablatian study on Double Mach dataset.

| Methods | | PDE-Diffusion | | -BC | | -PDE guidance | | - LRP | |
|---|---|---|---|---|---|---|---|---|---|
| Metric | | MSE | MAE | MSE | MAE | MSE | MAE | MSE | MAE |
| Double Mach | 2 | **0.113** | **0.135** | 0.253 | 0.275 | 0.213 | 0.269 | 0.244 | 0.287 |
| | 8 | **0.336** | **0.367** | 0.477 | 0.578 | 0.434 | 0.487 | 0.553 | 0.508 |
| | 16 | **0.409** | **0.288** | 0.767 | 0.645 | 0.732 | 0.541 | 0.687 | 0.454 |

# 6 CONCLUSION

In this paper, we present PDE-Diffusion (Physics Embedded Diffusion Model), a two-stage computational model specifically designed for solving partial differential equations (PDEs). Built on a generic and efficient physics-embedding framework, PDE-Diffusion is capable of leveraging prior physical knowledge inherent in PDE systems. When evaluated on the PDEBenchmarks dataset, PDE-Diffusion demonstrates state-of-the-art performance.

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

## A DETAILED PROOF

Recall that the reverse process $p_{\theta_{DM}}(d_n^i|d_n^{i+1})$ is modeled as a Gaussian distribution $\mathcal{N}(\mu, \Sigma)$, thus we have :

$$\log p_{\theta_{DM}}(d_n^i \mid d_n^{i+1}) = -\frac{1}{2}\left(d_n^i - \mu\right)^T \Sigma^{-1}\left(d_n^i - \mu\right) + C. \tag{13}$$

Here we use the same (Dhariwal & Nichol, 2021a), we make assumption that $\log p_{\theta_{DM}}(d_n^i|d_n^{i+1})$ has low curvature compared to $\Sigma^{-1}$, which is reasonable when reasonable in the limit of infinite diffusion steps. Therefore we have the Taylor expansion around $z_n = \mu$:

$$\log p_\phi(e|d_n^i) \approx \log p_\phi(e|d_n^i) \mid_{d_n^i=\mu} + (d_n^i - \mu)\nabla_{d_n^i} \log p_\phi(e|d_n^i) \mid_{d_n^i=\mu} \tag{14}$$

$$= -2(d_n^i - \mu)\epsilon_\phi(d_n^i) + C_1, \tag{15}$$

where $C_1$ is a constant.

$$\log\left(p_{\theta_{DM}}(d_n^i|d_n^{i+1})p_\phi(e|d_n^i)\right) \approx -\frac{1}{2}(d_n^i - \mu)^T\Sigma^{-1}(d_n^i - \mu) - 2(\epsilon_\phi(d_n^i) - \mu)\epsilon_\phi + C_2$$

$$= -\frac{1}{2}(d_n^i - \mu + 2\Sigma\epsilon_\phi)^T\Sigma^{-1}(d_n^i - \mu + 2\Sigma\epsilon_\phi) + 2\epsilon_\phi^T\Sigma\epsilon_\phi + C_2$$

$$= -\frac{1}{2}(d_n^i - \mu + 2\Sigma\epsilon_\phi)^T\Sigma^{-1}(d_n^i - \mu + 2\Sigma\epsilon_\phi) + C_3$$

$$= \log p(z) + C_4, z \sim \mathcal{N}(\mu - 2\Sigma\epsilon_\phi, \Sigma). \tag{16}$$

The constant $C_4$ can be safely absorbed by the normalizing term $Z$ in equation 5. We have thus proved that the conditional transition operator can be optimized by shifting its mean by $-2\Sigma\epsilon_\phi$.

---

**Algorithm 3** DDIM Sampling: generate physical field from Gaussian noise

---
1: $d_N \sim \mathcal{N}(\mathbf{0}, \mathbf{I})$
2: **for** $n = N, N-1, \ldots, 1$ **do**           ▷ Sampling loop
3:      $\epsilon \sim \mathcal{N}(\mathbf{0}, \mathbf{I})$ if $t > 1$, else $\epsilon = 0$          ▷ Sample noise term
4:      Update $d_n$ to using the following expression:          ▷ Update latent residual $d_n$

$$\boldsymbol{x}_{t-1} = \sqrt{\alpha_{t-1}}\left(\frac{\boldsymbol{x}_t - \sqrt{1-\alpha_t}\epsilon_\theta^{(t)}(\boldsymbol{x}_t)}{\sqrt{\alpha_t}}\right) + \sqrt{1 - \alpha_{t-1} - \sigma_t^2} \cdot \epsilon_\theta^{(t)}(\boldsymbol{x}_t) + \sigma_t\epsilon_t.$$

5: **end for**

---

## B DATASET DESCRIPTION

The PDE-Diffusion mainly focus on the unsteady 2-dimensional case with specified PDE equations. PDEBench (Takamoto et al., 2023a) provides five 2D PDE dataset, among witch only 2 datasets (CFD2D, DR2D) provide data of all the physical variable appeared in the equations. To comprehensively evaluate the performance of PDE-Diffusion, we generated two new fluid datasets using WENO5 CFD scheme. Here we provides a detailed descriptions of the four dataset.

RMI and Double Mach are two common test cases in the field of fluid dynamics They both satisfy the two-dimensional Euler equation:

$$\frac{\partial \mathbf{U}}{\partial t} + \frac{\partial \mathbf{F}(\mathbf{U})}{\partial x} + \frac{\partial \mathbf{G}(\mathbf{U})}{\partial y} = 0,$$

where $\mathbf{U} = (\rho, \rho u, \rho v, E)^T, \mathbf{F}(\mathbf{U}) = \left[\rho u, \rho u^2 + p, \rho uv, (E+p)u\right]^T$ and $\mathbf{G}(\mathbf{U}) = \left[\rho v, \rho vu, \rho v^2 + p, (E+p)v\right]^T$. Here $\rho, u, v, p$ are respectively the fluid's density, horizontal velocity, vertical velocity, and pressure. The momentum density is $\rho\mathbf{v} = (\rho u, \rho v)$, and the total energy density is $E = \rho e + \rho|\mathbf{v}|^2/2$. To close this set of equations, the ideal-gas equation of state $p = (\gamma - 1)\rho e$, with a constant $\gamma = 1.4$ is used.

### B.1 RICHTMEYR-MESHKOV INSTABILITY

Under the instantaneous impact of shock waves, the fluid interface with initial small disturbances will rapidly become unstable, and as the disturbance continues to grow, the large-scale structure on the interface will gradually break down, and the flow field will eventually enter a turbulent mixing state. This shock induced interface instability phenomenon is called Richtmyer Meshkov (RM) instability. The evolution process of RM instability is very complex, involving many fundamental fluid mechanics issues such as shock wave dynamics, vortex dynamics, multiphase flow, turbulent transition, and turbulent mixing, and has important academic research value.

In our article, the RM problem is initialized with the following conditions :

$$(\rho, u, v, p) = \begin{cases} (1.4112, -\dfrac{665}{1556}, 0, 1.628), & x \geq 3.2 \\ (5.04, 0, 0, 1), & x \leq x0 \\ (1, 0, 0, 1), & 3.2 \geq x \geq x0, \end{cases} \tag{17}$$

where X0 is a mixed interface of two fluids with different densities, defined as follows:

$$x0 = 2.9 - 0.1 sin(2\pi y + 0.25) \tag{18}$$

This loading can be seen as a shock wave impacting the interface at Mach 2.4 on one side of the heavy fluid. In the calculation, we take the grid size of $[256, 256]$, and a time span of five seconds. Open boundary conditions are taken for the left and right sides of the flow field, while symmetric boundary conditions are taken for the upper and lower sides.

### B.2 DOUBLE MACH PROBLEM

The Double Mach reflection problem was suggested by Woodward and Colella as a standard test case for Euler solvers and is widely used in testing CFD algorithms.At the initial moment, there is a tilted shock wave on the left side of the flow field, with a Mach number of 10. The definition of flow field is as follows:

$$(\rho, u, v, p) = \begin{cases} (8, 8.25\dfrac{\sqrt{3}}{2}, \dfrac{-8.25}{2}, 116.5), & x \leq x0 \\ (1.4, 0, 0, 1), & x \geq x0, \end{cases} \tag{19}$$

where x0 is the shock wave interface, defined as follows:

$$x0 = \frac{1}{6} + \sqrt{3}y \tag{20}$$

In the calculation, we take the grid size of $[256, 256]$, and the calculation time is $t = 0.3$. Open boundary conditions are taken for the upper,left and right sides of the flow field, while symmetric boundary conditions are taken for the lower sides

### B.3 CFD2D

The 2DCFD dataset is originally provided in PDEBench and contains 11 generated datasets based on 11 different PDEs. The 2DCFD dataset describes time-dependent compressible flow in a 2D space, which complies with the compressible Navier-Stokes equations:

$$\partial_t \rho + \nabla \cdot (\rho \mathbf{v}) = 0 \tag{21}$$

$$\rho \left( \partial_t \mathbf{v} + \mathbf{v} \cdot \nabla \mathbf{v} \right) = -\nabla p + \eta \Delta \mathbf{v} + \left( \zeta + \frac{\eta}{3} \right) \nabla(\nabla \cdot \mathbf{v}) \tag{22}$$

$$\partial_t \left[ \epsilon + \frac{\rho v^2}{2} \right] + \nabla \cdot \left[ \left( \epsilon + p + \frac{\rho v^2}{2} \right) \mathbf{v} - \mathbf{v} \cdot \sigma' \right] = 0, \tag{23}$$

where $\mathbf{v} = (u_x, u_v)^T$ is the 2D velocity vector, and $\epsilon = 1.5p$ is the internal energy. The dataset adopts an outgoing boundary condition that allows waves and fluids to escape from the computational domain (Stone & Norman, 1992). The 2DCFD dataset includes two cases of initial conditions. The random field initial condition is given by:

$$u^0(x, t_0) = \sum_{k_i = k_1, \dots, k_N} A_i \sin\left(k_i x + \phi_i\right), \tag{24}$$

where $k_i$ are the wave numbers defined as $k_i = \frac{2\pi n_i}{L_x}$, $n_i$ are random integers in the range $[1, n_{\max}]$, $L_x$ represents the size of the computational domain, $A_i$ are random floating-point numbers chosen uniformly from the range $[0, 1]$, and $\phi_i$ are randomly selected phases in the range $(0, 2\pi)$.

The turbulence initial conditions are similar to those given by Equation equation 24, with the only difference being that the coefficient $A_i = \frac{\bar{v}}{|k|^d}$.

### B.4 DR2D

PDEBench provides detailed description of the diffusion-reaction datase. Here we provide a brief overview of the basic information about the dataset to facilitate readers' quick understanding. The PDEs of the dataset are as follows:

$$\partial_t u = D_u \nabla^2 u + R_u(u, v)$$
$$\partial_t v = D_v \nabla^2 v + R_v(u, v),$$

where $D_u$ and $D_v$ represent the diffusion coefficients for the activator and inhibitor, and $R_u(u, v)$ and $R_v(u, v)$ are their respective reaction functions. The simulation domain spans $x \in (-1, 1)$, $y \in (-1, 1)$, and $t \in (0, 5]$, making it particularly suitable for modeling biological pattern formation.

The Fitzhugh-Nagumo equations define the reaction functions:

$$R_u(u, v) = u - u^3 - k - v$$
$$R_v(u, v) = u - v$$

With $k = 5 \times 10^{-3}$, and diffusion coefficients $D_u = 1 \times 10^{-3}$ and $D_v = 5 \times 10^{-3}$. The initial condition is generated as standard normal random noise $u(0, x, y) \sim \mathcal{N}(0, 1.0)$ for $x \in (-1, 1)$ and $y \in (-1, 1)$.

