# OpenReview forum: "PDE-Diffusion: Physic guided diffusion model for solving partial derivative equations"
_ICLR.cc/2024/Conference — Submitted to ICLR 2024_

### Official Review · Reviewer_Hhh3 · 2023-10-28

**Soundness:** 1 poor
**Presentation:** 2 fair
**Contribution:** 1 poor
**Rating:** 3
**Confidence:** 4

**Summary:**

This paper aims to leverage a diffusion-based model for solving Partial Differential Equations (PDEs), targeting improvements in generalizability, interpretability, and long-term predictive performance. Although the proposed PDE_Diffusion model exhibits some improvement in error metrics compared to other deep learning-based approaches, the paper lacks a rigorous justification for why diffusion models are well-suited for solving deterministic PDEs and how it should work.

I think the paper suffers from various technical and presentation issues. I recommend rejecting the paper for the following reasons: (1) the methodology is poorly elucidated, accompanied by ambiguous notations and questionable mathematical arguments, and (2) it fails to provide a compelling rationale for employing diffusion models, which are inherently probabilistic, in solving deterministic PDEs.

**Strengths:**

The paper demonstrates through experiments that diffusion models may hold promise for solving PDEs.

**Weaknesses:**

Major Comments:

1. Diffusion models are conventionally used to model distributions, focusing on generating samples that are similar to the training dataset. In contrast, this paper focuses on a deterministic task: solving a well-defined PDE. How do diffusion models contribute to solving such deterministic problems? Is the sample generated by the reverse SDE consistently close to the deterministic ground truth with small variance? If so, can one replace the reverse SDE with a deterministic process and initial conditions to obtain deterministic and improved results?

2. Although I am familiar with PDEs and diffusion models, I find the methodology section difficult to follow due to unclear notation and inexact mathematical expressions. There are many problems, I just list a few.

  (a) In the first paragraph of section 4.2, $\mathbf{D}$ denotes a decoder. According to $\mathbf{D}(z^0, d^i)$, $d^i$ should be a single snapshot in the latent space as residual. However, the next line states $d^i = (d^1_1, \dots, d^i_N)$, consisting of all the intermediate steps in the diffusion process.

  (b) What is the meaning of "probability that $d_m^i$ conforms to the governing equations $e$"?

  (c) Equations (5)(6)(7)(8) appear to model a conditional diffusion model. However, it is known in the literature that there is no straightforward way to model a conditional diffusion model directly from the original diffusion model without certain approximations (see e.g., the discussion in (Song et al., 2020b)). Since the paper does not clarify the meaning of conditional on $e$, it is unclear what this section is attempting to convey. Equation (6) seems more like an assumption than a mathematical derivation.

  (d) Equation (6) involves the notation $\hat{u}^{i-1}_n$ and $\hat{u}^{i}_n$, but the next line discusses the notation $\hat{u}^{n}_i$ and $\hat{u}^{n}_{i-1}$.

  (e) What is meant by "derivative kernel" above equation (12)?

3. The paper claims that solving some PDEs takes "months-long simulations utilizing iterative numerical solvers." However, I believe the PDEs tested in this paper are far from such a regime. Solving these PDEs should be relatively quick, perhaps just a few minutes. In contrast, each physical-time evolution step in the proposed PDE-Diffusion model involves multiple diffusion steps, which can be time-consuming, not to mention the extensive training time required for these diffusion models. The authors should provide the solving time of classical solvers, inference time per solution, and total training time for diffusion models to give readers a sense of the time cost.

4. There are placeholders in the paper where results should be reported. For instance, on page 7, it states "PDE-Diffusion has a decrease of x% on average in the PRE metric" and "Our method shows significantly better results in the case of longer prediction window sizes. The MSE decrease is x% (at 2), x% (at 4), x% (at 8), x% (at 16)." These "x%" placeholders indicate hasty composition, suggesting the authors did not even thoroughly proofread the paper.

5. The paper cites (Gao et al., 2022) when introducing the significance of solving PDEs in the first sentence. However, this paper has a very loose connection with PDEs. A keyword search for "PDE" in that paper reveals that the term only appears twice in the related work section. Citing such a work (Gao et al., 2022) with minimal relevance to the paper's focus on PDEs, particularly in the opening sentence, is quite unprofessional.

Minor Issues:

1. What is the "PRE metric"? The paper lacks a definition.

2. In the problem definition, the authors present solving a PDE as finding an approximate solution on a fixed spatial discretization. This is misleading. In the field of numerical PDEs, the choice of spatial discretization is integral to the PDE-solving process, and most numerical methods can accommodate varying discretizations. The authors should clarify that their method is not mesh-free.

3. Typo in the appendix: "which is reasonable when reasonable in the limit of infinite diffusion steps".

4. In several instances, the authors refer to "partial derivative equations." This term is unconventional; Google search results suggest it is rarely used.

**Questions:**

See questions above.

---

### Official Review · Reviewer_t73C · 2023-10-30

**Soundness:** 2 fair
**Presentation:** 2 fair
**Contribution:** 2 fair
**Rating:** 3
**Confidence:** 3

**Summary:**

This paper proposes a diffusion-based model, called PDE-Diffusion, to solve partial differential equations. This method is a two-stage model consisting of an autoencoder and a diffusion model. PDE-Diffusion incorporates physics-based priors to enhance model interpretability and generalization. Moreover, the model assimilates PDE-informed constraints to ensure temporal coherence while producing high quality predictions.

**Strengths:**

- The proposed method exhibits better interpretability and generalizability by embedding physical constraints into the reverse process of diffusion model.
- The method adopts a latent residual to mitigate the problem of temporal incoherence in physical field predictions.
- This paper introduce two new datasets that could enhance the available resources for using deep learning to solve PDEs.

**Weaknesses:**

- There’s other diffusion-based PDE solver that should be compared to, such as DYffusion [1]
- The baseline is not enough. FNO is not a recent method and not the state-of-the-art model. I recommend authors to compare with more recent neural PDE solvers to better evaluate the effectiveness of the proposed method, such as FFNO [2] and GFNO [3].
- In most rows in table 2 and 3, the MSE and MAE of PDE-Diffusion is larger than the other baselines but is still annotated boldface, which is misleading.
- In section 5.3, many placeholders are not replaced by final results.


[1] Cachay, Salva Rühling, et al. "DYffusion: A Dynamics-informed Diffusion Model for Spatiotemporal Forecasting." arXiv preprint arXiv:2306.01984 (2023).\
[2] Tran, Alasdair, et al. "Factorized fourier neural operators." arXiv preprint arXiv:2111.13802 (2021). \
[3] Helwig, Jacob, et al. "Group Equivariant Fourier Neural Operators for Partial Differential Equations." arXiv preprint arXiv:2306.05697 (2023).

**Questions:**

- What is the motivation of using diffusion model to solve PDE and advantages over autoregressive methods?
- The proposed method seems not use autoregressive or operator way to predict solution, instead this method adopt N markov chains that corresponds to N time steps. Then how to predict time step that is not seen during the training?

---

### Official Review · Reviewer_p93J · 2023-11-01

**Soundness:** 3 good
**Presentation:** 1 poor
**Contribution:** 2 fair
**Rating:** 3
**Confidence:** 5

**Summary:**

To address the challenge of interpretability, generalizability and long-horizon predictive performance, the author proposed the PDE-Diffusion model. The model incorporate physics-based priors, two stage diffusion model to tackle the mentioned challenges. It was also tested on extensive datasets.

**Strengths:**

Originality: The author propose to embed a physics-based prior into the sampling process for diffusion model to enhance the learning result.

Quality and clarify: The methodology part misses details about the physics-based priors. The result part is confusing, indicating the proposed model cannot beat the SOTA models.

Significance: The author claims that the physics-embedded framework is more interoperable and generalizable, meanwhile mitigate the problem of temporal coherence.

**Weaknesses:**

The methodology part is not clear in cases of how to embed the physics prior $\epsilon_I$ and $\epsilon_B$ during the reverse sampling process. The result doens't support the claim where in many cases, the PDE-Diffusion result is much worse than FNO. The reviewer also has major concern about the writing style. It seems not rigorous and scientific on delivering the message. Details can be found in the questions part.

**Questions:**

1. Why not put the algorithm 3 in the main manuscript? The innovative part physics-conditioning is in algorithm 3 while algorithm 1 and 2 are well-known algorithm. Moreover, I didn't see the equation on how to deal with the $\epsilon_I$ and $\epsilon_B$ in either main manuscript or the appendix. How do you embed the physics prior into the model?

2. The result is confusing. In Table 2 and 3, the result for the diffusion models are exactly the same. However, the metrics are evaluated for different datasets. How is it possible? Moreover, the bold font seems to be used pretty arbitrary, not always the best result. In fact, the proposed model's performance is worse than FNO in may cases. In that case, the result doesn't support the authors' claim that the proposed structure is better than SOTA.

3. The writing is poor, seems not to be proofread. To name a few. In Table 2, for FNO, there is a blank without numbers. In page 7 section 5.3, "has a decrease of x%" is obviously not a completed part. The authors should fill the exact number instead of a placeholder. In page 13, "which is reasonable when reasonable in the ..." is obviously a wrong sentence.

---

### Official Review · Reviewer_Mvr7 · 2023-11-03

**Soundness:** 1 poor
**Presentation:** 2 fair
**Contribution:** 2 fair
**Rating:** 1
**Confidence:** 4

**Summary:**

This paper introduce a diffusion-based method for simulating PDEs. The method performs diffusion in the latent space, and incorporate physics priors into the model. Experiments are conducted in various number of datasets in PDE-Bench and 2 introduced datasets.

**Strengths:**

Significance: The paper addresses the important problem of improving the generalizability and long-horizon predictive performance of simulating PDEs. The diffusion has advantage compared to prior approaches.

Novelty: The introduction of latent diffusion to PDE simulation, to my knowledge, is novel.

Clarity: The introduction of the method is mostly clear.

**Weaknesses:**

My most concern with the paper is the soundness of the paper, since I find that there are many errors in the tables in the experiment section.

Errors:
1. The most obvious is in Table 2 and Table 3, where the PDE diffusion has exactly THE SAME results for all the different datasets, and even the same results in Table 2 and Table 3. This is clearly impossible.

2. Table 2 lacks the results for dataset CFD2D turbo2 for FNO, with MAE metric.

3. in Table 1, the bold font should make the best result across all methods. However, CFD2D turbo2, the proposed method is clearly has higher MAE and MSE than FNO.

Other places:
1. in Table 1, DR2D lacks the bold font for any method for 2-step.

2. In section 5.3, in the third last row, it lacks the numbers: "The MSE decrease is x% (at 2), x% (at 4), x%..."

3. In Fig 2, "Double shock" should be "Double Mach"?

These errors and unfinished parts seems be due to the time rush, so the paper is not polished enough. I strongly suggest the authors to double and triple check the paper, to make sure everything is correct, before submitting. I believe that if the paper is properly and carefully done and well-polished, it can be a strong paper. In its current form, I have to suggest rejection.

Besides, it would be great if the authors can analyze why the proposed method outperforms VDM. A comparison with existing work DYffusion [1] may also be preferable. There are also many neural PDE works that evolves the system in latent space, e.g., [2][3][4][5][6]. It may be preferable to discuss them.

[1] DYffusion: A Dynamics-informed Diffusion Model for Spatiotemporal Forecasting, NeurIPS 2023

[2] Multiscale simulations of complex systems by learning their effective dynamics, Nature Machine Intelligence

[3] Learning to Accelerate Partial Differential Equations via Latent Global Evolution, NeurIPS 2022

[4] Latent space subdivision: stable and controllable time predictions for fluid flow, in Computer Graphics Forum

[5] Model reduction of dynamical systems on nonlinear manifolds using deep convolutional autoencoders,” Journal of Computational Physics

[6] Deep fluids: A generative network for parameterized fluid simulations,” in Computer Graphics Forum

**Questions:**

N/A

---

### Official Review · Reviewer_uZRL · 2023-11-07

**Soundness:** 1 poor
**Presentation:** 1 poor
**Contribution:** 1 poor
**Rating:** 1
**Confidence:** 5

**Summary:**

The paper considers a two-stage diffusion-based model for solving partial differential equations (PDE). It conducts experiments on 4 benchmark datasets to compare its performance with some leading methods.

**Strengths:**

The paper explores a diffusion-based method for solving partial differential equations. It compares the diffusion-based method performance with some leading methods on four benchmark datasets.

**Weaknesses:**

1. The manuscript is essentially unfinished. There are many typos, missing figures in the table (e.g. Table 2. FNO MAE CFD2D turb2), missing values (e.g. 5.3 Results Analysis: The MSE decrease is x% (at 2), x% (at 4)...).

2. The comparative results show that FNO is better than the proposed. Table 3 shows that except for one or two cases, the PEDM performs worse compared to FNO. In Table 3, the average of the six random initializations with variance should have been reported rather than the individual results of the initialization.

3. The paper fails to compare with more advanced techniques than FNO.

4. The motivation for the use of the diffusion model for solving PDE is not sound. Diffusion is founded on a probabilistic setting while PDE is deterministic and it isn't easy to relate the two.

5. The inference time comparison between the different methods should be shown in comparison. FNO finds the solution in a parallel manner while PEDM does not.

**Questions:**

Please show the inference time to compare the methods.

**Details Of Ethics Concerns:**

None.

---

### Meta-Review · Area_Chair_uhQo · 2023-12-06

**Metareview:**

The paper proposes PDE-Diffusion, a new approach to solve PDEs using diffusion models, along with two new datasets for benchmarking model performance. The reviewers pointed out several shortcomings  (including the numerous typos and missing results, the lack of sufficient comparisons, and questionable benefits over the current state-of-the-art), resulting in a unanimous reject.

**Justification For Why Not Higher Score:**

This was a unanimous reject.

**Justification For Why Not Lower Score:**

N/A

---

### Decision · Program_Chairs · 2024-01-16

Reject